# STAPS : Training-Free Zero-Shot Anomaly Detection via Semantic-Temporal Scoring and Prototype Selection

## Abstract

Zero-shot anomaly detection (ZAD) addresses the need for anomaly detection without large-scale labeled datasets by leveraging large pretrained representations without domain-specific supervision. However, existing ZAD methods still depend on labeled pretraining, limiting their applicability in practical scenarios. Training-free ZAD eliminates this dependency by directly leveraging pretrained backbones without additional training, offering a cost-efficient alternative. However, training-free ZAD suffers from semantic bias by applying class-oriented representations to anomaly detection without fine-tuning. In this work, we propose a novel training-free framework Semantic-Temporal scoring and Prototype Selection (STAPS) that mitigates semantic bias and incorporates temporal context into anomaly detection. The proposed method comprises two key components. First, semantic-temporal anomaly scoring refines anomaly scores that are biased toward class semantics by leveraging temporal locality and continuity to capture sequential dependencies. Second, bayesian gaussian mixture-based prototype selection automatically identifies prototypes sensitive to anomaly evidence, thereby reducing semantic bias in backbone features and enhancing pixel-level anomaly segmentation. Extensive experiments on nine benchmark datasets demonstrate that our proposed method achieves state-of-the-art performance, achieving 91.9% image-level AUROC for anomaly detection and 97.7% pixel-level AUROC for anomaly segmentation, highlighting both robustness and generalizability.

## 1 Introduction

Traditional anomaly detection methods primarily rely on learning from normal data to distinguish anomalies. These methods typically model the distributional characteristics of normal data or predefined features, and then classify samples that deviate from this distribution as anomalies. However, such approaches (Zhang et al., 2023; He et al., 2024; Guo et al., 2025) require large-scale normal datasets and incur high labeling costs, while failing to adequately capture the diversity and unpredictability of data in real-world applications. To address these limitations, Zero-shot Anomaly Detection (ZAD) emerged. ZAD directly leverages the representational power of large-scale pretrained models, enabling anomaly detection without an explicit training process. The goal of ZAD is to generalize to new domains and detect anomalies without domain-specific supervision. Despite these advances, existing ZAD methods still rely on pretraining using labeled datasets, limiting their applicability in real-world situations.

As shown in Fig. 1, the training-free ZAD methods perform anomaly detection and classification by leveraging only the inherent generalization capacity of the pre-trained backbone network without requiring additional training. This approach is cost-effective and has low dependence on normal patterns during the training process. Among them, MuSc (Li et al., 2024) is attracting attention as an efficient method for simultaneously detecting and classifying anomalies by comparing samples within a batch using only pre-trained representations. However, MuSc directly applies the class semantic representation of the pre-trained backbone to anomaly detection without any fine-tuning or adaptation. As shown in Fig. 2, this causes the model overly sensitive to semantic class differences, regardless of whether a defect is present. The lack of a downstream fine-tuning process prevents correction of such semantic bias, constituting a fundamental challenge that restricts anomaly detec-

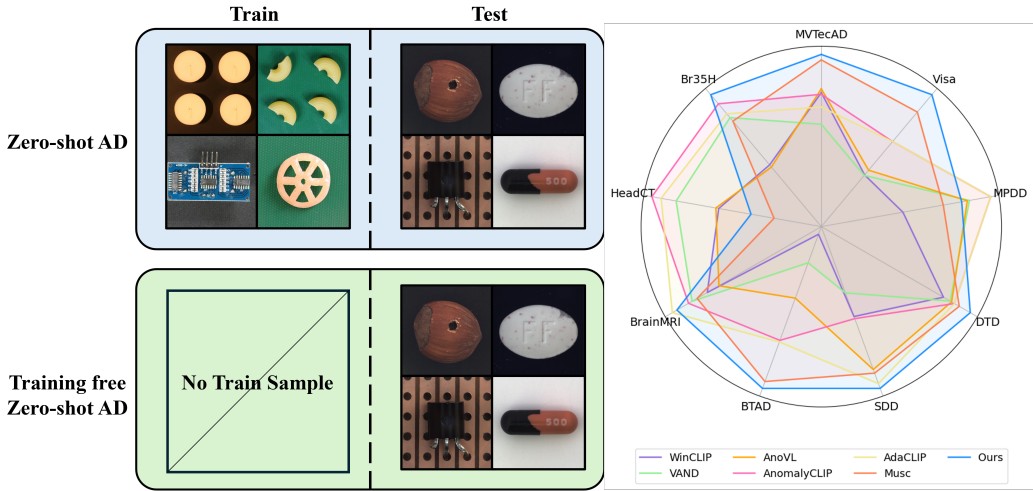

Figure 1: Left: Illustrations for target and auxiliary dataset of zero-shot, and training-free zero-shot anomaly detection paradigms. Right: Quantitative comparison with popular methods by image-level AUROC on industrial and medical datasets.

tion performance. To address these limitations, we draw inspiration from treating an image dataset not merely as a static collection but as a video sequence endowed with temporal attributes. In this setting, abnormal regions become more evident when observing transitions across consecutive frames, exhibiting a noticeable discrepancy compared to normal regions. Motivated by this observation, we propose Semantic-Temporal Anomaly Scoring (STAS), which improves anomaly scores initially dominated by class semantics by incorporating temporal locality and continuity to capture the sequential context required in real-world applications.

Fig. 2 compares the nearest-neighbor retrieval results between MuSc, which relies solely on semantic similarity for anomaly scoring, and our method, which additionally incorporates temporal consistency. For each input image, both methods select the top-3 neighbors based on their respective similarity measures. In case (a), when the input is a hazelnut image with a painting defect, MuSc incorrectly selects an image with a different defect type (cut), whereas our method accurately retrieves only painting defect images. In case (b), MuSc selects the semantically closest image to the input, while our method selects normal images of the same object but with different shapes, thereby better distinguishing normal and abnormal conditions. In cases (c) and (d), MuSc incorrectly selects abnormal neighbors for normal input images, whereas in this study, only normal samples are consistently selected. These results suggest that MuSc remains biased toward class semantics, which tends to confuse normal and abnormal states. In contrast, in this study, temporal relationships are utilized to mitigate semantic bias and select neighbors that more accurately reflect the true condition of the input image. Furthermore, we propose Bayesian Gaussian Mixture-based Prototype Selection (BGMPS) to automatically identify anomaly-sensitive prototypes among multiple candidates. BGMPS encourages the model to focus on anomaly-relevant features rather than class distinctions, thereby alleviating semantic bias in the backbone representation. Finally, by leveraging the cosine similarity between the selected prototypes and other clusters, our method highlights anomalous pixels, which serve as anomaly evidence, and further improves pixel-level anomaly segmentation performance.

We validated the effectiveness of our approach by evaluating both anomaly detection and anomaly segmentation through extensive experiments on nine diverse benchmark datasets. As shown in Fig. 1, our approach consistently achieves state-of-the-art performance across all tasks, demonstrating strong robustness and generalization. In particular, our method achieves an image-level AUROC of 91.9% for anomaly detection and a pixel-level AUROC of 97.7% for anomaly segmentation. Our key contributions are summarized as follows:

- We propose STAPS, the first training-free zero-shot anomaly detection framework that assigns temporal attributes to image datasets, enabling anomaly detection beyond static semantic representations.

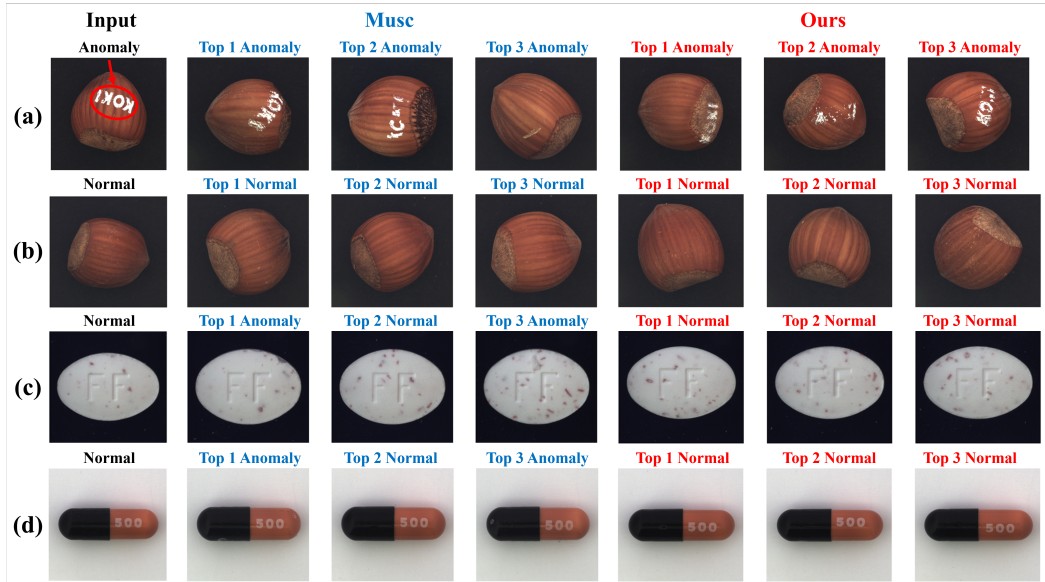

Figure 2: Qualitative comparison of nearest-neighbor retrieval based on different similarity measures. For each input image, the top row in blue (MuSc) shows the top-3 neighbors retrieved using semantic similarity only, while the bottom row in red (Ours) shows the neighbors retrieved using the proposed semantic and temporal similarity.

- We introduce Semantic-Temporal Anomaly Scoring (STAS), which treats an image dataset as a video-like sequence to move beyond static class semantics. By integrating temporal locality and continuity, STAS mitigates semantic bias and enables anomaly detection that transitions from static semantic similarity to context-aware temporal reasoning.

- We propose Bayesian Gaussian Mixture-based Prototype Selection (BGMPS), which leverages prototype selection based on anomaly evidence to emphasize anomaly-relevant features, suppress semantic bias, and enable more precise pixel-level anomaly segmentation, thereby introducing a new perspective in anomaly detection.

- We conduct extensive experiments on nine diverse benchmark datasets, demonstrating that our method achieves state-of-the-art performance in both anomaly detection and segmentation, with 91.9% image-level AUROC and 97.7% pixel-level AUROC.

## 2 RELATED WORKS

### 2.1 ZERO-SHOT ANOMALY DETECTION

Due to the difficulty of collecting abnormal data, traditional anomaly detection methods (You et al., 2022; Lu et al., 2023; Zhang et al., 2023; He et al., 2024; Guo et al., 2025) have primarily focused on modeling the distribution of normal data or exploiting reconstruction errors. However, these approaches often require large-scale normal datasets and expensive annotations, limiting their generalizability in real-world applications. To address these limitations, Zero-shot Anomaly Detection (ZAD) has been proposed as a paradigm that leverages large pretrained representation models to distinguish normality and abnormality in auxiliary datasets without additional training on the target domain. Recent studies (Radford et al., 2021) have explored the use of vision-language models, where visual features are aligned with semantic concepts through textual prompts. For example, CLIP-based methods employ class labels or prompt engineering to separate normal from abnormal samples, while subsequent studies such as such as WinCLIP (Jeong et al., 2023), AdaCLIP (Cao et al., 2024), APRIL-GAN (Chen et al., 2023), AnomalyCLIP (Zhou et al., 2023) and AA-CLIP (Ma et al., 2025) have introduced more elaborate prompt designs and adaptive mechanisms to im-

prove detection performance. Nevertheless, these methods remain constrained by semantic bias inherent in pretraining and have limited robustness to domain variations.

## 2.2 TRAINING-FREE ZERO-SHOT ANOMALY DETECTION

To alleviate the reliance on additional training or prompt engineering, recent advances have focused on training-free Zero-shot Anomaly Detection (ZAD) approaches (Jeong et al., 2023; Cao et al., 2024; Chen et al., 2023; Zhou et al., 2023; Ma et al., 2025). Training-free ZAD directly exploits the representational capacity of large pretrained models to identify anomalies without parameter optimization or fine-tuning, making this paradigm particularly practical within zero-shot learning. For instance, MuSc (Li et al., 2024) distinguishes anomalies by comparing samples within a batch based solely on their relative feature distances, while other studies propose anomaly detection by computing the similarity between input samples and textual prompts such as "normal" in the embedding space of pretrained Vision Transformers (ViTs) (Dosovitskiy et al., 2020) or CLIP. In this way, training-free ZAD leverages the generalization capabilities of large-scale pre-trained models while eliminating the need to collect and train on normal data, thereby enabling real-time and cost-effective anomaly detection. Furthermore, this approach can be readily adapted to distributional shifts and new categories, which enhances applicability in real-world scenarios.

## 2.3 BAYESIAN GAUSSIAN MIXTURE MODEL

The Bayesian Gaussian Mixture Model (BGMM) (Blei and Jordan, 2006) provides a probabilistic framework for clustering that automatically adapts the number of effective components. Unlike traditional KMeans (Arthur and Vassilvitskii, 2006) or GMM (Dempster et al., 1977), BGMM incorporates Bayesian prior probabilities, enabling more robust estimation under uncertainty and preventing overfitting to abnormal clusters. In the context of anomaly detection, BGMM has been employed to identify prototypes or cluster structures that emphasize anomaly-related features, improving discriminability while reducing sensitivity to class semantics. Their ability to capture heterogeneous data distributions makes them particularly suitable for zero-shot settings.

# 3 METHOD

## 3.1 OVERVIEW

This paper proposes STAPS, a novel training-free framework for ZAD that effectively mitigates the semantic bias of pretrained backbones. As shown in Fig. 3, our method integrates two new modules based on MuSc, leveraging both temporal cues and prototype selection. First, we introduce Semantic-Temporal Anomaly Scoring (STAS), which treats an image dataset as a video-like sequence (not actual temporal data, but a pseudo-temporal ordering that imposes locality and continuity) and improves anomaly scores by leveraging temporal coherence. This allows defects to be identified not only from spatial irregularities but also from inconsistent behavior across consecutive pseudo-frames. Second, we propose Bayesian Gaussian Mixture-based Prototype Selection (BGMPS), which clusters intermediate anomaly maps with DCT-based embedding and bayesian gaussian mixture modeling to automatically select prototypes that are sensitive to anomaly evidence. This alleviates semantic biases inherited from pretrained representations and emphasizes anomaly-related pixels through prototype fusion based on cosine similarity. By jointly applying STAS and BGMPS, our framework produces refined anomaly maps with improved localization and achieves robust detection across diverse domains.

## 3.2 SEMANTIC-TEMPORAL ANOMALY SCORING (STAS)

Conventional training-free approaches often generate anomaly scores dominated by class semantics. To alleviate this bias, we propose Semantic-Temporal Anomaly Scoring (STAS), which improves anomaly scores by integrating semantic similarity and temporal continuity.

**Semantic matrix** Given $N$ test images, let $X \in \mathbb{R}^{N \times D}$ denote the global embeddings (CLS tokens) extracted from a pre-trained backbone. We first normalize each embedding to unit length and compute the cosine similarity between all pairs, forming a semantic similarity matrix $S \in \mathbb{R}^{N \times N}$.

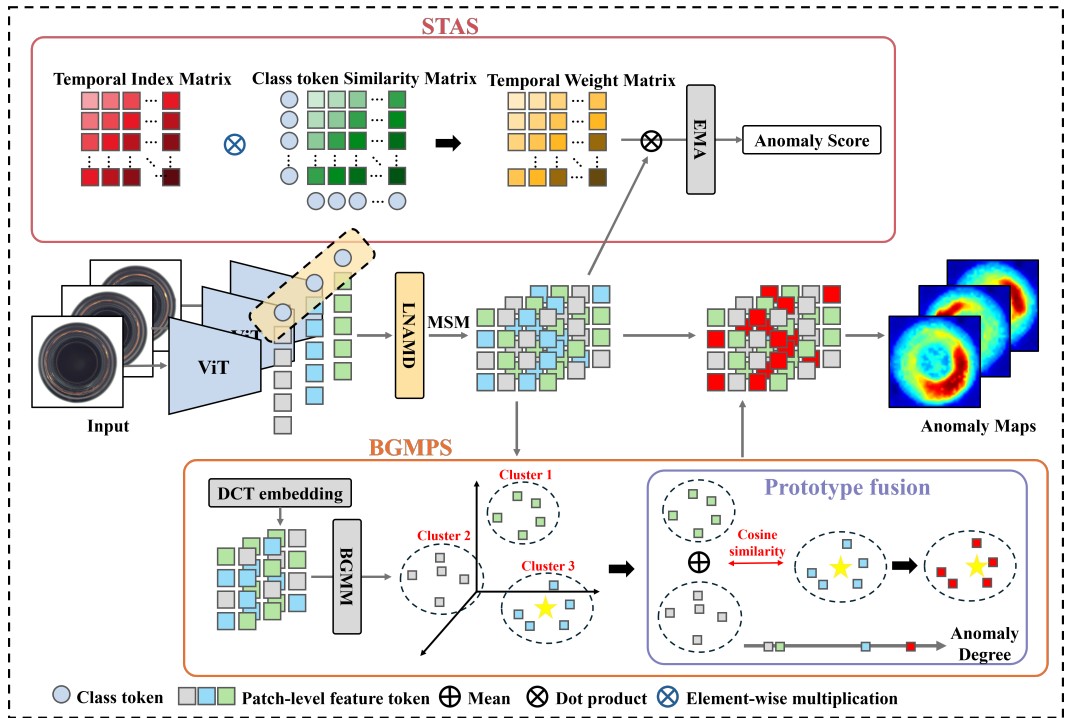

Figure 3: Framework of STAPS. The model extracts visual tokens and class tokens from a visual encoder. LNAMD and MSM compute patch-wise similarities to generate anomaly maps, which are then refined by BGMPS and prototype fusion to emphasize anomaly-relevant patches. Finally, anomaly scores are derived by the STAS module, which incorporates a temporal weight matrix and EMA smoothing.

This matrix is further rescaled into the range $[0, 1]$, such that $S_{ij}$ represents the semantic affinity between images $i$ and $j$.

**Temporal matrix** We then construct a temporal adjacency matrix $T \in \mathbb{R}^{N \times N}$ to encode the continuity of class token embeddings along the sequence, assigning higher weights to temporally adjacent images. Specifically, for two images $i$ and $j$, the weight is defined as

$$T_{ij} = \begin{cases} \gamma^{|i-j|}, & |i-j| \leq L, \\ 0, & \text{otherwise,} \end{cases} \tag{1}$$

where $L$ is the temporal window size and $\gamma \in (0, 1)$ is an exponential decay factor. This ensures that only neighbors within $\pm L$ steps contribute, with the weight decreasing as the temporal distance $|i - j|$ increases.

**Score refinement** The two matrix are combined into a weighted adjacency matrix:

$$W = \alpha S + (1 - \alpha)T, \tag{2}$$

where $\alpha$ balances the relative influence of the semantic and temporal graphs, and $W$ represents the combined adjacency matrix. Given the initial anomaly scores $s \in \mathbb{R}^N$ computed by MSM, we refine them by propagating scores across $W$. For each image $i$, we select the top-$k$ neighbors with the strongest edge weights $W_{i\ell}$, normalize these weights row-wise, and then take the weighted average of their scores. To improve robustness, this procedure is repeated for multiple values of $k$, and the results are averaged to produce the final refined score $\hat{s}_i$. In this way, each score is adjusted according to both semantically similar and temporally adjacent neighbors, suppressing noisy semantic effects and highlighting anomalies that disrupt temporal consistency.

**Temporal smoothing**   Finally, we apply Exponential Moving Average (EMA) (Hunter, 1986) smoothing along the temporal sequence to further stabilize fluctuations:

$$\hat{s}_t^{\text{EMA}} = \begin{cases} \hat{s}_t, & t = 0, \\ \beta \hat{s}_{t-1}^{\text{EMA}} + (1-\beta)\hat{s}_t, & t > 0, \end{cases} \tag{3}$$

where $\beta \in [0,1)$ controls the degree of smoothing, and $t$ represents the time interval of the sequence. This time interval reduces score variations over time and produces more stable temporal predictions.

By integrating semantic similarity, temporal locality, and EMA smoothing, STAS mitigates semantic bias and improves the robustness of training-free ZAD.

### 3.3 BAYESIAN GAUSSIAN MIXTURE-BASED PROTOTYPE SELECTION (BGMPS)

**Prototype selection.**   While STAS mitigates the instability of image-level scoring by incorporating temporal information, patch-level anomaly localization still suffers from semantic bias in the backbone representations. To address this issue, we propose Bayesian Gaussian Mixture-based Prototype Selection (BGMPS), which automatically selects prototypes that are sensitive to anomaly evidence. Given $E$ anomaly maps obtained from different aggregation degrees or backbone stages, represented as $maps \in \mathbb{R}^{E \times N \times P}$, where $P$ is the number of pixels per anomaly map, each individual anomaly map is called an expert. Each map is vectorized into $X \in \mathbb{R}^{E \times (N \cdot P)}$ and normalized. A Discrete Cosine Transform (DCT) (Ahmed et al., 2006) is then applied to obtain a compact embedding $Z \in \mathbb{R}^{E \times d}$, where $d$ is the embedding dimension after DCT compression, that preserves dominant structural information. We cluster these embeddings using a Bayesian Gaussian Mixture Model (BGMM), which partitions the experts into prototype groups. For each cluster $c$, we compute the mean anomaly map $\overline{M}_c$ and evaluate the evidence score by averaging the top $p\%$ of pixel values in $\overline{M}_c$. The cluster with the highest evidence score is selected as the anomaly-sensitive prototype cluster.

**Prototype fusion.**   Let $C \in \mathbb{R}^{E_c \times N \times P}$ represent the anomaly maps of the selected cluster with $E_c$ experts. We compute the centroid $\overline{C}$, which is the average of all experts, and measure the cosine similarity between each expert map and the centroid to obtain the confidence weights $w \in \mathbb{R}^{E_c}$. Negative similarities are clipped to zero and the weights are normalized to sum to one. The final fused anomaly map is then given by

$$M_{\text{fused}} = \sum_{e=1}^{E_c} w_e C_e, \tag{4}$$

where $C_e$ represents the anomaly map of the $e$-th expert. $M_{\text{fused}}$ represents the final fused anomaly map that integrates anomaly-sensitive prototypes into a single refined representation, serving as the final attention map for pixel-level anomaly localization. This procedure emphasizes experts that are more consistent with the centroid while suppressing those with noise or inconsistencies. By explicitly selecting prototypes according to anomaly evidence and fusing them in a similarity-based manner, BGMPS effectively reduces semantic bias and improves pixel-level anomaly localization.

## 4 EXPERIMENTS

### 4.1 EXPERIMENTAL SETUPS

To evaluate our STAPS, we conducted experiments on nine benchmark datasets. These included MVTecAD (Bergmann et al., 2021), Visa (Zou et al., 2022), MPDD (Jezek et al., 2021), DTD (Aota et al., 2023), SDD (Tabernik et al., 2020), and BTAD (Mishra et al., 2021) for the industrial domain, as well as BrainMRI (Kanade and Gumaste, 2015), HeadCT (Kitamura, 2018), and Br35H (Hamada, 2020) for the medical domain. As the backbone network, we use ViT-L/14-336 pretrained by OpenAI (Radford et al., 2021), from which patch token features were extracted at the 5th, 11th, 17th, and 23rd layers. All input images from the industrial domain datasets were resized to $518 \times 518$ pixel resolution, while images from the medical domain datasets were resized to $256 \times 256$ pixel resolution before being fed into the model. Inference was conducted using a single NVIDIA RTX 4090 GPU, with a batch size of 4. As the primary evaluation metric, we adopted the Area Under

Table 1: Comparison of ZAD methods with image-level AUROC metric. Bold and underlining indicate best results and second-best results, respectively.

| Method | Industrial domain | | | | | | Medical domain | | | Average |
|---|---|---|---|---|---|---|---|---|---|---|
| | MVTecAD | Visa | MPDD | DTD | SDD | BTAD | BrainMRI | HeadCT | Br35H | |
| WinCLIP | 91.8 | 78.1 | 63.6 | 93.2 | 84.3 | 68.2 | 86.6 | 81.8 | 80.5 | 80.9 |
| APRIL-GAN | 86.1 | 78.0 | 73.0 | 94.6 | 79.8 | 73.6 | 89.6 | 89.2 | 91.4 | 83.9 |
| AnoVL | 92.5 | 79.2 | 72.7 | 94.9 | 94.4 | 80.3 | 84.3 | 82.3 | 80.0 | 84.5 |
| AnomalyCLIP | 91.5 | 85.8 | **76.0** | 95.5 | 84.7 | 88.3 | 90.3 | **93.4** | 94.6 | 86.7 |
| AdaCLIP | 89.2 | 85.8 | **76.0** | 95.5 | 97.1 | 88.6 | **93.5** | 91.8 | 92.3 | 90.0 |
| Musc | 97.7 | 92.6 | 69.3 | 96.5 | 95.1 | 96.1 | 85.7 | 76.9 | 85.2 | 88.3 |
| Ours | **98.7** | **96.6** | 71.9 | **98.8** | **98.0** | **97.4** | 91.0 | 82.6 | **94.7** | **92.0** |

Table 2: Comparison of ZAD methods with pixel-level AUROC metric. Bold and underlining indicate best results and second-best results, respectively.

| Method | Industrial domain | | | | | | Medical domain | | | Average |
|---|---|---|---|---|---|---|---|---|---|---|
| | MVTecAD | Visa | MPDD | DTD | SDD | BTAD | BrainMRI | HeadCT | Br35H | |
| WinCLIP | 85.1 | 79.6 | 76.4 | 83.9 | 68.8 | 72.7 | - | - | - | 77.8 |
| APRIL-GAN | 87.6 | 94.2 | 94.1 | 95.3 | 79.8 | 60.8 | - | - | - | 85.3 |
| AnoVL | 90.6 | 85.2 | 62.3 | 97.7 | 97.1 | 75.2 | - | - | - | 84.7 |
| AnomalyCLIP | 91.1 | 95.5 | 96.5 | **97.9** | 90.6 | 94.2 | - | - | - | 94.3 |
| AdaCLIP | 88.7 | 95.5 | 96.1 | **97.9** | **97.7** | 92.1 | - | - | - | 94.7 |
| Musc | 97.1 | **98.7** | **97.4** | 97.6 | 97.0 | 97.4 | - | - | - | 97.5 |
| Ours | **98.0** | 98.4 | **97.4** | 97.7 | 96.6 | **98.1** | - | - | - | **97.7** |

the Receiver Operating Characteristic (AUROC), which is widely regarded as a robust measure for anomaly detection. AUROC evaluates the trade-off between true positive and false positive rates across thresholds, making it threshold-independent. In anomaly detection, this provides a reliable basis for assessing both image-level anomaly identification and pixel-level anomaly localization.

## 4.2 COMPARISON WITH ZERO-SHOT METHODS

We compared our framework against two categories of ZAD approaches. For training-free ZAD methods, we selected WinCLIP, AnoVL, and MuSc, while APRIL-GAN, AnomalyCLIP, and Ada-CLIP were chosen as training-required ZAD baselines. Table 1 reports the image-level AUROC comparison across both industrial and medical domains. Overall, our method achieves the highest average performance among all competing approaches, consistently surpassing existing baselines in both domains.

In the industrial domain, our framework consistently outperforms prior approaches. For instance, it records the highest AUROC on MVTecAD (98.7%), Visa (96.6%), DTD (98.8%), SDD (98.0%), and BTAD (97.4%), while maintaining competitive results on MPDD (71.9%). Although some existing methods, such as AdaCLIP, exhibit competitive scores on certain datasets, our approach delivers the most stable improvements across diverse benchmarks. In the medical domain, our method also demonstrates clear advantages over existing ZAD approaches. On BrainMRI, it achieves an AU-ROC of 92.6%, outperforming most baselines and ranking second only to AdaCLIP. On Br35H, our framework secures the best result with 96.7%, substantially surpassing AnomalyCLIP (94.6%) and AdaCLIP (92.3%). Although the performance on HeadCT (76.2%) is lower than that of Anomaly-CLIP and AdaCLIP, our method still maintains competitive results without dataset-specific training or fine-tuning. Notably, the consistent improvements on BrainMRI and Br35H confirm that the proposed STAS enhances robustness even in complex and heterogeneous medical imaging scenarios.

As shown in Tables 2, our method achieves the best overall performance with an average AUROC of 97.7%, consistently surpassing strong recent baselines such as AnomalyCLIP and AdaCLIP. In particular, significant gains are observed on large-scale datasets such as Visa (98.4%) and BTAD (98.1%), further validating the effectiveness of our framework under challenging distributional set-

Table 3: Ablation study on the contributions of STAS, BGMPS, and EMA with image, pixel-level AUROC. Bold indicate best results.

| Method | MVTecAD | Visa | MPDD | DTD | SDD | BTAD | Average |
|---|---|---|---|---|---|---|---|
| Musc | 97.7, 97.1 | 92.6, 97.1 | 69.3, **97.4** | 96.5, 97.6 | 95.1, **97.0** | **97.4**, 97.4 | 91.4, 97.3 |
| w/o STAS | 96.8, **98.0** | 91.1, 98.4 | 70.8, **97.4** | **98.9**, 97.7 | 95.1, 96.6 | 96.0, **98.1** | 91.5, **97.7** |
| w/o BGMPS | **98.7**, 97.3 | **96.6**, **98.6** | **71.9**, **97.4** | 98.8, 97.5 | **98.0**, 96.9 | **97.4**, 97.4 | **93.6**, 97.5 |
| w/o EMA | 98.2, **98.0** | 95.2, 98.4 | **71.9**, **97.4** | 97.7, **97.7** | 96.2, 96.6 | 97.1, **98.1** | 92.7, **97.7** |
| Full Model | **98.7**, **98.0** | **96.6**, 98.4 | **71.9**, **97.4** | 98.8, **97.7** | **98.0**, 96.6 | **97.4**, **98.1** | **93.6**, **97.7** |

Table 4: Ablation study on the effect of EMA parameter $\beta$ with image-level AUROC. Bold indicate best results.

| EMA | MVTecAD | Visa | MPDD | DTD | SDD | BTAD | Average |
|---|---|---|---|---|---|---|---|
| 0 | 98.2 | 95.2 | **71.9** | 97.7 | 96.2 | 97.1 | 92.7 |
| 0.1 | 98.3 | 95.6 | **71.9** | 97.9 | 96.6 | 97.4 | 93.0 |
| 0.3 | 98.5 | 96.1 | 71.3 | 98.4 | 97.4 | **97.5** | 93.2 |
| 0.5 (current) | **98.7** | **96.6** | **71.9** | **98.8** | **98.0** | 97.4 | **93.6** |

tings. Overall, these results demonstrate that the proposed method establishes a new state-of-the-art in training-free ZAD, while consistently outperforming CLIP-based training-required baselines across both industrial and medical domains.

## 4.3 EFFECTIVENESS OF STAS, BGMPS, AND EMA

To evaluate the contributions of each component in our framework, we conducted an ablation study on STAS, BGMPS, and EMA, with results summarized in Table 3.

Removing STAS leads to a significant drop in image-level AUROC, highlighting the importance of semantic-temporal aggregation for robust scoring. By incorporating a temporal index matrix into semantic similarity based scoring, STAS effectively assigns temporal properties to inter-image relationships, thereby enhancing the stability and reliability of anomaly detection.

Removing BGMPS consistently decreases pixel-level AUROC across all datasets, demonstrating its crucial role in alleviating semantic bias and enabling prototypes to better capture abnormal features. In particular, BGMPS leverages clustering to guide normal samples toward prototypes with high normality and anomalous samples toward prototypes with high abnormality, thereby achieving a clearer separation between normality and anomaly. Without this process, the detector tends to overfit to class semantics rather than anomaly discrimination, resulting in degraded generalization.

Finally, removing EMA slightly reduces stability in image-level AUROC, with the effect being more pronounced on datasets with higher variance such as Visa and SDD. This shows that EMA smoothing, which has traditionally been applied only to time series data, can also operate effectively in image datasets without explicit temporal attributes by implicitly assigning temporal continuity across adjacent images. In other words, EMA mitigates temporal variations and forms more stable decision boundaries, demonstrating the potential to exploit temporal context even in non-sequential image data.

Overall, the full model achieves the best balance, validating that the proposed components are complementary and collectively contribute to the robustness and generalization ability of our training-free anomaly detection framework.

## 4.4 EFFECT OF EMA PARAMETER

To examine the role of temporal smoothing, we conduct an ablation study by varying the EMA parameter $\beta$, as shown in Table 4. When $\beta = 0$, i.e., without exponential moving average, the model

Table 5: Comparison of different embedding and clustering combinations on MVTecAD and Visa datasets with pixel-level AUROC.

| DCT | SVD | KMeans | BGMM | MVTecAD | Visa |
|:---:|:---:|:---:|:---:|:---:|:---:|
| ✓ | ✗ | ✗ | ✓ | **98.0** | **98.4** |
| ✓ | ✗ | ✓ | ✗ | **98.0** | 98.1 |
| ✗ | ✓ | ✓ | ✗ | **98.0** | 97.7 |
| ✗ | ✓ | ✗ | ✓ | **98.0** | 98.1 |

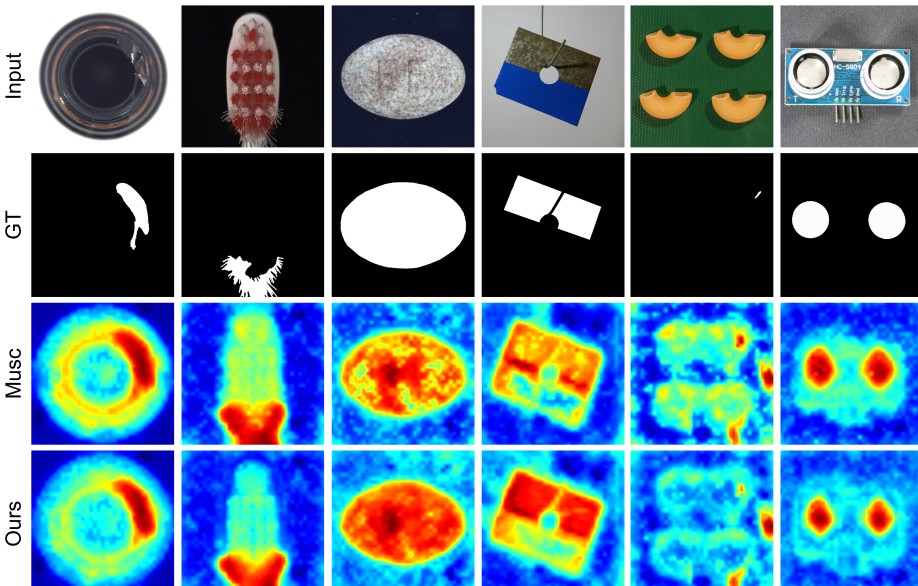

Figure 4: Qualitative results for anomaly localization on various domain datasets. From top to bottom: anomalous sample, ground-truth, predicted anomaly maps from Musc, and our predicted anomaly map.

already achieves strong performance, but the scores exhibit slight instability across datasets. Introducing a small degree of smoothing ($\beta = 0.1$) improves overall consistency, and further increasing the weight ($\beta = 0.3$) leads to additional gains, especially on datasets with larger intra-class variance such as Visa and SDD.

The best results are obtained with $\beta = 0.5$, which provides the highest average AUROC (93.6%). This indicates that EMA effectively suppresses frame-wise fluctuations in anomaly scores while preserving discriminability. Notably, excessively low $\beta$ fails to fully mitigate noise, whereas higher values strike a favorable balance between stability and sensitivity. These results highlight the importance of temporal smoothing in our framework, complementing STAS by enhancing robustness under diverse conditions.

### 4.5 QUALITATIVE ANALYSIS OF ANOMALY LOCALIZATION

Figure 4 presents qualitative comparisons between Musc and our method across various domains. Musc frequently highlights semantically similar but anomaly-irrelevant regions, reflecting its sensitivity to class semantics. In contrast, our method produces more coherent and defect-focused anomaly maps. This behavior is attributed to two factors: (1) STAS suppresses semantically inconsistent responses by enforcing locality-aware score refinement, and (2) BGMPS selects anomaly-relevant prototypes, reducing reliance on semantic similarity. The proposed framework consistently

Table 6: Effect of permutation on STAS.

| Permutations | Image-level Avg | Pixel-level Avg |
|---|---|---|
| 1 permutation | 92.2 | 97.7 |
| 4 permutations | 92.4 | 97.7 |
| 7 permutations | 89.9 | 97.6 |

emphasizes true defect regions while suppressing false activations, demonstrating clearer spatial precision and reduced semantic bias.

### 4.6 EMBEDDING AND CLUSTERING CHOICES IN BGMPS

We further analyze the design choices of BGMPS by comparing different embedding and clustering strategies, as summarized in Table 5. For embedding, both DCT and SVD (Golub and Reinsch, 1971) yield competitive results, consistently achieving 98% AUROC on MVTecAD. However, on Visa, DCT-based embedding produces slightly higher scores (98.4%) compared to SVD (97.7–98.1%), suggesting that DCT provides a more compact yet effective representation for capturing anomaly-sensitive variations.

Regarding clustering, both KMeans and BGMM perform reliably, but BGMM generally yields marginally better stability, particularly when combined with DCT. The probabilistic nature of BGMM enables it to better adapt to the heterogeneous distributions of anomaly evidence, while KMeans, though simpler, remains competitive with only a slight performance drop.

These results indicate that while our framework is robust across different embedding and clustering configurations, the combination of DCT embedding with BGMM clustering provides the most consistent performance, validating our default choice in BGMPS.

## 5 EFFECT OF PSEUDO-TEMPORAL ORDERING

The pseudo-temporal axis in STAS does not correspond to real temporal information; instead, it is constructed from an arbitrary shuffle of the test set using a fixed random seed. To verify that the method does not rely on any specific ordering, we evaluated STAS under multiple input permutations.

As shown in Table 7 and Table 8, changing the random seed produces nearly identical results across all datasets. In addition, Table 6 reports the effect of repeatedly permuting the entire test set one, four, and seven times. Both image-level and pixel-level AUROC remain stable under these perturbations. These results demonstrate that the pseudo-temporal adjacency acts as a lightweight locality regularizer rather than encoding meaningful temporal structure. STAS is therefore *ordering-invariant*, showing robustness to both random shuffling and strong permutation disruption of the test set.

## 6 CONCLUSION

This paper presented STAPS, a training-free framework for zero-shot anomaly detection that integrates semantic-temporal anomaly scoring and prototype selection. By combining STAS, BGMPS, and EMA, the proposed method alleviates semantic bias, enhances anomaly-sensitive representations, and improves score stability. Extensive experiments on nine industrial and medical benchmarks demonstrate that our method consistently achieves state-of-the-art performance in both image-level anomaly detection and pixel-level anomaly segmentation, confirming the robustness and generalizability of the approach. Although STAPS shows strong performance across diverse datasets, BGMPS improves pixel-level localization by selecting anomaly-sensitive prototypes. However, in cases where the data distribution is highly heterogeneous, its performance may vary depending on the quality of clustering. Addressing this limitation remains an important direction for future work.

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

# SUPPLEMENTARY MATERIAL

## A    USE OF LARGE LANGUAGE MODELS

We used large language models (LLMs) exclusively for grammar correction and minor typographical editing of the manuscript.

## B    EFFECT OF RANDOM SEED

To further verify the robustness of our method, we evaluate the performance under different random seeds. As shown in Table 7 and Table 8, the results remain stable across all datasets and evaluation metrics, demonstrating that our method is insensitive to random initialization and produces consistent results.

Table 7: Image-level AUROC under different random seeds.

| Seed | MVTec | Visa | MPDD | DTD | SDD | BTAD | Brain MRI | Br35H | HeadCT | Average |
|---|---|---|---|---|---|---|---|---|---|---|
| 1234 | 98.7 | 96.6 | 70.2 | 98.8 | 98.0 | 97.4 | 92.6 | 96.7 | 76.2 | 91.7 |
| 1111 | 98.7 | 96.6 | 70.2 | 98.8 | 98.0 | 97.4 | 92.6 | 96.7 | 76.2 | 91.7 |
| 0 | 98.7 | 96.6 | 70.2 | 98.8 | 98.0 | 97.4 | 92.6 | 96.7 | 76.2 | 91.7 |
| 42 | 98.7 | 96.6 | 71.9 | 98.8 | 98.0 | 97.4 | 92.6 | 96.7 | 76.2 | 91.9 |

Table 8: Pixel-level AUROC under different random seeds.

| Seed | MVTec | Visa | MPDD | DTD | SDD | BTAD | Brain MRI | Br35H | HeadCT | Average |
|---|---|---|---|---|---|---|---|---|---|---|
| 1234 | 97.9 | 98.1 | 97.4 | 97.7 | 96.6 | 98.0 | – | – | – | 97.6 |
| 1111 | 97.9 | 98.1 | 97.4 | 97.7 | 96.6 | 98.2 | – | – | – | 97.7 |
| 0 | 98.0 | 98.1 | 97.4 | 97.8 | 96.6 | 98.2 | – | – | – | 97.7 |
| 42 | 98.0 | 98.4 | 97.4 | 97.7 | 96.6 | 98.1 | – | – | – | 97.7 |

Table 9: Performance on the MVTecAD dataset. Each entry reports image-level (AUROC, F1, AP) and pixel-level (AUROC, F1, AP, AUPRO).

| Category | Image-level | | | Pixel-level | | | |
|---|---|---|---|---|---|---|---|
| | AUROC | F1 | AP | AUROC | F1 | AP | AUPRO |
| bottle | 100.0 | 100.0 | 100.0 | 99.13 | 81.17 | 86.55 | 96.67 |
| cable | 100.0 | 100.0 | 100.0 | 98.85 | 72.34 | 73.51 | 93.03 |
| capsule | 99.24 | 98.62 | 99.84 | 98.44 | 39.63 | 39.68 | 93.99 |
| carpet | 99.84 | 98.89 | 99.95 | 99.54 | 75.22 | 80.65 | 97.83 |
| grid | 100.0 | 100.0 | 100.0 | 98.92 | 47.25 | 42.74 | 95.17 |
| hazelnut | 100.0 | 100.0 | 100.0 | 99.21 | 71.51 | 72.12 | 91.83 |
| leather | 99.93 | 99.46 | 99.98 | 99.74 | 64.56 | 66.70 | 98.76 |
| metal_nut | 100.0 | 100.0 | 100.0 | 90.32 | 60.02 | 59.99 | 92.38 |
| pill | 99.70 | 99.29 | 99.94 | 97.90 | 68.75 | 74.46 | 97.84 |
| screw | 83.34 | 86.35 | 93.94 | 97.68 | 27.93 | 19.98 | 90.52 |
| tile | 99.93 | 99.40 | 99.97 | 99.01 | 81.55 | 87.13 | 95.97 |
| toothbrush | 100.0 | 100.0 | 100.0 | 99.44 | 68.79 | 61.17 | 93.45 |
| transistor | 99.04 | 96.39 | 98.28 | 95.77 | 64.94 | 67.60 | 87.51 |
| wood | 99.82 | 99.16 | 99.95 | 98.09 | 70.11 | 77.74 | 95.29 |
| zipper | 99.13 | 98.76 | 99.74 | 98.28 | 60.85 | 59.80 | 94.66 |
| Mean | 98.67 | 98.42 | 99.44 | 98.02 | 63.64 | 64.65 | 94.33 |

Table 10: Performance on the VisA dataset. Each entry reports image-level (AUROC, F1, AP) and pixel-level (AUROC, F1, AP, AUPRO).

| Category | Image-level | | | Pixel-level | | | |
|---|---|---|---|---|---|---|---|
| | AUROC | F1 | AP | AUROC | F1 | AP | AUPRO |
| cashew | 100.0 | 100.0 | 100.0 | 99.29 | 74.69 | 77.35 | 90.44 |
| macaroni2 | 70.32 | 68.04 | 77.54 | 95.33 | 9.28 | 2.74 | 82.35 |
| candle | 99.29 | 96.62 | 99.26 | 99.31 | 41.37 | 31.65 | 96.67 |
| capsules | 97.10 | 95.92 | 98.68 | 98.43 | 50.71 | 44.62 | 87.30 |
| chewinggum | 100.0 | 100.0 | 100.0 | 99.03 | 54.72 | 49.44 | 87.53 |
| fryum | 100.0 | 100.0 | 100.0 | 97.36 | 55.26 | 47.74 | 93.14 |
| macaroni1 | 97.22 | 93.81 | 98.06 | 97.95 | 14.79 | 6.93 | 88.85 |
| pcb1 | 95.71 | 91.37 | 96.43 | 99.10 | 76.17 | 81.39 | 83.23 |
| pcb2 | 99.51 | 96.52 | 99.50 | 97.94 | 32.57 | 19.69 | 83.86 |
| pcb3 | 100.0 | 100.0 | 100.0 | 98.81 | 33.37 | 30.77 | 91.74 |
| pcb4 | 99.99 | 99.50 | 99.99 | 99.01 | 53.05 | 53.77 | 93.42 |
| pipe_fryum | 100.0 | 100.0 | 100.0 | 99.44 | 71.22 | 70.93 | 97.39 |
| Mean | 96.60 | 95.15 | 97.46 | 98.41 | 47.27 | 43.08 | 89.66 |

Table 11: Performance on the BTAD dataset. Each entry reports image-level (AUROC, F1, AP) and pixel-level (AUROC, F1, AP, AUPRO).

| Category | Image-level | | | Pixel-level | | | |
|---|---|---|---|---|---|---|---|
| | AUROC | F1 | AP | AUROC | F1 | AP | AUPRO |
| 1 | 99.81 | 98.97 | 99.92 | 97.97 | 61.39 | 61.63 | 85.91 |
| 2 | 92.25 | 95.92 | 98.66 | 97.10 | 67.58 | 72.92 | 69.62 |
| 3 | 99.99 | 98.80 | 99.88 | 99.35 | 56.34 | 59.03 | 97.30 |
| Mean | 97.35 | 97.90 | 99.49 | 98.14 | 61.77 | 64.52 | 84.28 |

Table 12: Performance on the SDD dataset (mean results). Each entry reports image-level (AUROC, F1, AP) and pixel-level (AUROC, F1, AP, AUPRO).

| Dataset | Image-level | | | Pixel-level | | | |
|---|---|---|---|---|---|---|---|
| | AUROC | F1 | AP | AUROC | F1 | AP | AUPRO |
| SDD (mean) | 98.03 | 91.26 | 94.55 | 96.58 | 35.28 | 25.94 | 92.82 |

Table 13: Performance on the DTD dataset. Each entry reports image-level (AUROC, F1, AP) and pixel-level (AUROC, F1, AP, AUPRO).

| Category | Image-level | | | Pixel-level | | | |
|---|---|---|---|---|---|---|---|
| | AUROC | F1 | AP | AUROC | F1 | AP | AUPRO |
| Woven_001 | 99.90 | 99.28 | 99.96 | 99.61 | 69.65 | 75.80 | 97.28 |
| Woven_127 | 99.77 | 99.38 | 99.75 | 93.17 | 64.18 | 63.99 | 92.16 |
| Stratified_154 | 97.88 | 96.86 | 99.51 | 99.75 | 78.89 | 86.82 | 98.48 |
| Blotchy_099 | 100.0 | 100.0 | 100.0 | 99.82 | 80.05 | 86.95 | 98.46 |
| Woven_068 | 99.70 | 98.09 | 99.81 | 98.34 | 63.50 | 65.78 | 95.37 |
| Woven_125 | 99.75 | 99.37 | 99.94 | 99.67 | 76.11 | 83.84 | 98.81 |
| Marbled_078 | 100.0 | 100.0 | 100.0 | 99.62 | 76.12 | 82.36 | 98.64 |
| Perforated_037 | 94.31 | 94.81 | 98.74 | 94.08 | 57.73 | 56.36 | 86.83 |
| Mesh_114 | 95.69 | 95.48 | 98.53 | 93.35 | 60.57 | 57.59 | 78.43 |
| Fibrous_183 | 100.0 | 100.0 | 100.0 | 99.68 | 78.42 | 86.21 | 98.71 |
| Matted_069 | 99.43 | 99.37 | 99.85 | 99.44 | 71.80 | 76.46 | 97.84 |
| Woven_104 | 98.63 | 99.38 | 99.61 | 96.38 | 65.19 | 67.30 | 91.61 |
| Mean | 98.75 | 98.50 | 99.64 | 97.74 | 70.18 | 74.12 | 94.38 |

Table 14: Performance on the MPDD dataset. Each entry reports image-level (AUROC, F1, AP) and pixel-level (AUROC, F1, AP, AUPRO).

| Category | Image-level | | | Pixel-level | | | |
|---|---|---|---|---|---|---|---|
| | AUROC | F1 | AP | AUROC | F1 | AP | AUPRO |
| bracket_black | 59.24 | 74.60 | 73.03 | 96.05 | 13.28 | 5.19 | 91.04 |
| bracket_brown | 55.88 | 79.69 | 73.21 | 94.28 | 8.19 | 3.84 | 90.50 |
| bracket_white | 23.89 | 66.67 | 36.51 | 98.21 | 0.99 | 0.25 | 93.04 |
| connector | 84.76 | 84.85 | 57.15 | 97.91 | 34.13 | 25.38 | 92.90 |
| metal_plate | 99.19 | 97.26 | 99.70 | 99.09 | 88.14 | 93.09 | 96.51 |
| tubes | 98.05 | 95.71 | 99.12 | 99.11 | 64.89 | 68.44 | 96.79 |
| Mean | 70.17 | 83.13 | 73.12 | 97.44 | 34.94 | 32.70 | 93.46 |

Table 15: Performance on the Br35h dataset (mean results).

| Dataset | AUROC | F1 | AP |
|---|---|---|---|
| Br35h (mean) | 96.66 | 93.75 | 94.19 |

Table 16: Performance on the BrainMRI dataset (mean results).

| Dataset | AUROC | F1 | AP |
|---|---|---|---|
| BrainMRI (mean) | 92.55 | 94.15 | 91.06 |

Table 17: Performance on the HeadCT dataset (mean results).

| Dataset | AUROC | F1 | AP |
|---|---|---|---|
| HeadCT (mean) | 76.15 | 77.05 | 68.35 |

