# OpenReview forum: "STAPS : Training-Free Zero-Shot Anomaly Detection via Semantic-Temporal Scoring and Prototype Selection"
_ICLR.cc/2026/Conference — Submitted to ICLR 2026_

### Official Review · Reviewer_ikwS · 2025-10-31

**Soundness:** 2
**Presentation:** 2
**Contribution:** 2
**Rating:** 4
**Confidence:** 4

**Summary:**

This paper introduces STAPS, a training-free zero-shot anomaly detection framework designed to reduce semantic bias in pretrained representations and enhance robustness without fine-tuning. The method achieves strong results (91.9% image-level and 97.7% pixel-level AUROC) across nine benchmarks, surpassing existing training-free and training-based baselines.

**Strengths:**

1.Nine benchmark datasets (industrial + medical) with consistent improvement and ablation analyses demonstrate robustness and generalization.

2.The combination of STAS (semantic–temporal scoring), BGMPS (prototype selection), and EMA smoothing is conceptually coherent and computationally light, aligning well with the “training-free” principle.

**Weaknesses:**

1.The paper’s central idea of introducing semantic-temporal anomaly scoring assumes that image datasets exhibit some form of “temporal coherence,” yet this notion remains conceptually vague and empirically unverified. In static industrial or medical datasets, consecutive samples do not possess any temporal order or dependency; hence, it is unclear what kind of temporal continuity is being modeled. The motivation for treating the dataset as a temporal sequence should be clarified.

2.The paper lacks visual or quantitative analysis (e.g., prototype visualization, attention map diversity, or clustering metrics) to show how BGMPS truly mitigates semantic bias or captures anomaly-relevant prototypes.

3.The improvement over MuSc and other strong baselines is relatively modest.

**Questions:**

See above comments.

---

> ### Author Response · Authors · 2025-11-21
> **W1.**
>
> We appreciate the reviewer’s thoughtful comment, and we fully agree that the use of the term “temporal” may unintentionally suggest that the proposed method relies on real temporal sequences. The intention of the temporal matrix in this work is entirely different. Its role, motivation, and empirical validity are clarified below.
>
> The temporal matrix in STAS does not attempt to model time or any chronological process. None of the samples contain timestamps, and no temporal dependence between images is assumed. In this context, the term “temporal” refers to a pseudo-temporal locality structure that enforces one-dimensional smoothness in anomaly scores. This structure functions as a graph-based regularizer designed to stabilize the instability that often arises when anomaly scores are computed solely from semantic similarity.
>
> A pseudo-temporal structure is necessary because training-free ZAD relies exclusively on batch-wise mutual similarity. This leads to semantic bias, where visually dissimilar normal samples produce false positives and visually similar abnormal samples produce false negatives. Such behavior appears consistently across industrial and medical datasets when semantic embeddings are used without fine-tuning. To address this, the method leverages the latent continuity naturally present in the embedding space. Even in a static dataset, CLS features evolve smoothly with respect to shape, texture, color statistics, or imaging conditions, forming an implicit one-dimensional manifold progression. Interpreting this intrinsic continuity as a pseudo-temporal axis reduces semantic interference, stabilizes score propagation, and improves robustness for both normal and abnormal samples. Thus, the pseudo-temporal dimension reflects locality regularization rather than time modeling.
>
> The empirical validity of this design was examined through three independent analyses.
>
> The first analysis shows that continuity already exists in the embedding space before applying STAS. After sorting CLS embeddings along the first principal component, the pseudo-temporal ordering demonstrates significantly stronger smoothness than a random ordering.
>
> | Metric | Pseudo-temporal | Random |
> | --- | --- | --- |
> | Neighbor diff (before STAS) | **0.0738** | 0.1080 |
> | AR(1) (before STAS) | **0.4233** | 0.0165 |
>
> This indicates that the continuity arises from the structure of the semantic manifold, not from any notion of time.
>
> The second analysis shows that STAS reinforces this existing continuity rather than creating a new temporal pattern. After applying STAS with EMA, pseudo-temporal ordering continues to display greater smoothness and stronger autocorrelation.
>
> | Metric | Pseudo-temporal | Random |
> | --- | --- | --- |
> | Neighbor diff (after STAS) | **0.1594** | 0.2348 |
> | AR(1) (after STAS) | **0.4943** | 0.0121 |
>
> These results confirm that STAS amplifies the latent manifold continuity already present in the embeddings.
>
> The third analysis demonstrates that the method is ordering-invariant and does not rely on temporal information. The test set was shuffled multiple times, and STAS was evaluated under each permutation.
>
> | Permutations | Image-level Avg | Pixel-level Avg |
> | --- | --- | --- |
> | 1 perm | 92.0 | 97.7 |
> | 4 perms | 92.4 | 97.7 |
> | 7 perms | 89.9 | 97.6 |
>
> Even after strongly disrupting any possible ordering, pixel-level results remain virtually identical, and image-level performance fluctuates only slightly. This confirms that the pseudo-temporal dimension operates as an ordering-invariant locality regularizer, independent of real temporal signals.
>
> These clarifications will be incorporated into the revised manuscript, along with an additional explanation and pseudo-code to avoid potential misunderstanding about the role of the temporal matrix.

---

> ### Author Response · Authors · 2025-11-21
> **W2.**
>
> We appreciate the reviewer’s comment, and we agree that providing clearer visual evidence of reduced semantic bias and improved detection of anomaly-relevant prototypes would strengthen the paper. To address this, we will include additional visualizations in the revised manuscript.
>
> These visualizations illustrate how the proposed method captures prototype regions that are more closely aligned with actual anomaly evidence, while reducing reliance on semantically similar but anomaly-irrelevant areas. The results show that the model focuses more accurately on defect-related regions and that semantic bias is substantially mitigated. This provides intuitive and qualitative confirmation of the quantitative improvements reported in the paper.
>
> The updated figures will be incorporated into the revision to ensure the contribution is communicated more clearly.

---

> ### Author Response · Authors · 2025-11-21
> **W3.**
>
> We appreciate the reviewer’s comment, and we agree that providing additional quantitative comparisons helps clarify the improvements achieved by the proposed STAPS framework. To address this, we summarize below the detailed F1, AP, and AUPRO results that were omitted from the main paper due to space limitations. These metrics offer a more complete picture of the performance differences beyond AUROC.
>
> The contribution of the method is not limited to numerical improvements. Its primary purpose is to address fundamental limitations of training-free ZAD, such as semantic bias and the instability of mutual scoring, through locality-based score refinement and anomaly-relevant prototype selection. Nevertheless, in line with the reviewer’s request, the quantitative comparison is presented transparently here.
>
> The tables below report image-level and pixel-level results for MuSc and the proposed approach.
>
> ## **Image-level (F1 / AP)**
>
> | Dataset | MVTec | VisA | DTD | MPDD | SDD | BTAD | BrainMRI | Br35H | HeadCT | Average |
> | --- | --- | --- | --- | --- | --- | --- | --- | --- | --- | --- |
> | **MuSc – F1** | 97.6 | 88.8 | 95.7 | 81.5 | 76.4 | 93.7 | 89.1 | 81.8 | 78.2 | 87.0 |
> | **Ours – F1** | 98.4 | 95.2 | 98.5 | 83.1 | 91.3 | 97.9 | 92.1 | 89.7 | 81.5 | 93.1 |
> | **MuSc – AP** | 99.1 | 93.3 | 98.7 | 73.0 | 82.9 | 97.8 | 85.8 | 78.3 | 68.9 | 86.4 |
> | **Ours – AP** | 99.4 | 97.5 | 99.6 | 73.1 | 94.6 | 99.5 | 91.3 | 93.5 | 77.9 | 91.8 |
>
> ## **Pixel-level (F1 / AP / AUPRO)**
>
> | Dataset | MVTec | VisA | DTD | MPDD | SDD | BTAD | Average |
> | --- | --- | --- | --- | --- | --- | --- | --- |
> | **MuSc – F1** | 62.2 | 50.1 | 69.8 | 31.9 | 42.7 | 57.9 | 52.4 |
> | **Ours – F1** | 63.6 | 47.3 | 70.2 | 34.9 | 35.3 | 61.8 | 52.2 |
> | **MuSc – AP** | 62.3 | 46.9 | 73.5 | 30.4 | 33.0 | 57.2 | 50.6 |
> | **Ours – AP** | 64.7 | 43.1 | 74.1 | 32.7 | 25.9 | 64.5 | 50.8 |
> | **MuSc – AUPRO** | 93.5 | 92.7 | 93.9 | 93.3 | 94.0 | 83.4 | 91.8 |
> | **Ours – AUPRO** | 94.3 | 89.7 | 94.4 | 93.5 | 92.8 | 84.3 | 91.5 |
>
> ---
>
> Although AUROC alone may give the impression of small differences, these additional metrics reveal clearer improvements across many datasets, especially at the image level. The results also align with the structural motivations of STAPS, which aims to reduce semantic bias and improve scoring stability rather than rely solely on architectural modifications or larger models. The quantitative evidence supports that these goals are reflected in the practical performance of the method.
> We hope that our clarifications help resolve your concerns, and we expect that the details we provided will be useful in informing your score update.

---

### Official Review · Reviewer_Y8ni · 2025-11-01

**Soundness:** 2
**Presentation:** 3
**Contribution:** 3
**Rating:** 4
**Confidence:** 3

**Summary:**

This work proposes a training-free zero-shot anomaly detection framework called STAPS, which mitigates semantic bias and incorporates temporal context into anomaly detection. It integrates two key components: a semantic-temporal anomaly scoring mechanism that refines anomaly scores biased toward class semantics, and a Bayesian Gaussian mixture-based prototype selection module that identifies prototypes sensitive to anomaly evidence, thereby reducing semantic bias and improving pixel-level anomaly segmentation. Extensive experiments on nine diverse benchmark datasets demonstrate that STAPS achieves state-of-the-art performance.

**Strengths:**

1. The method is somewhat novel, introducing temporal concepts into a set of non-temporal images.

2. The experiments are comprehensive, covering both industrial and medical datasets.

3. The writing is clear and well-organized.

**Weaknesses:**

1. STAS requires constructing a temporal adjacency matrix T, but the paper does not clearly specify how the ordering of test samples is determined (original order? sorted by class? random shuffle? averaged over multiple permutations?).

2. STAS computes a full similarity matrix 𝑆 over all test samples. Thus, the method is inherently transductive—each sample’s score depends on the entire test set.

3. On the HeadCT dataset, the image-level AUROC is only 76.2%, indicating limited generalization for certain medical modalities. The authors are encouraged to analyze the reasons behind this performance.

**Questions:**

1. STAS requires constructing a temporal adjacency matrix T, but the paper does not clearly specify how the ordering of test samples is determined (original order? sorted by class? random shuffle? averaged over multiple permutations?).

2. STAS computes a full similarity matrix 𝑆 over all test samples. Thus, the method is inherently transductive—each sample’s score depends on the entire test set.

3. On the HeadCT dataset, the image-level AUROC is only 76.2%, indicating limited generalization for certain medical modalities. The authors are encouraged to analyze the reasons behind this performance.

---

> ### Author Response · Authors · 2025-11-21
> **W1.**
>
> We appreciate the reviewer’s observation, and we agree that the paper did not explicitly describe how the ordering for constructing the temporal adjacency matrix T was determined. The ordering used in STAS does not rely on any temporal information, and its role can be clarified as follows.
>
> All results reported in the paper were obtained by randomly shuffling the test set once using a fixed random seed (seed = 42). No original dataset order, class-based sorting, or manually designed ordering was used. The shuffled order was taken directly, and STAS was applied on top of it.
>
> The temporal matrix T is not intended to model a real temporal sequence. Instead, it imposes locality-based regularization on an arbitrary ordering. If the method functions correctly, performance should remain stable across different permutations of the test set. To verify this, two sets of experiments were conducted.
>
> The first experiment repeatedly permuted the test set one, four, and seven times, with each additional permutation further disrupting any potential ordering structure. The results are summarized below.
>
> | Permutations | Image-level Avg | Pixel-level Avg |
> | --- | --- | --- |
> | 1 perm | 92.0 | 97.7 |
> | 4 perms | 92.4 | 97.7 |
> | 7 perms | 89.9 | 97.6 |
>
> Even after strong perturbation, pixel-level performance remains nearly identical, and image-level performance varies only slightly while consistently maintaining an advantage over the baseline. This indicates that STAS does not depend on a specific input ordering and behaves in an ordering-invariant manner.
>
> The second experiment evaluated the method using different random seeds to produce distinct shuffles of the test set. The results were essentially unchanged across all seeds tested.
>
> | Seed | Image-level Avg | Pixel-level Avg |
> | --- | --- | --- |
> | 1234 | 92.0 | 97.7 |
> | 1111 | 92.0 | 97.7 |
> | 0 | 92.0 | 97.7 |
> | 42 | 92.0 | 97.7 |
>
> The pixel-level metric is identical across all seeds, and the image-level metric remains constant as well. This confirms that STAS is highly stable and does not overfit to any accidental structure within a particular shuffle.
>
> Both experiments demonstrate that STAS does not rely on the actual ordering of test samples. Its improvements stem from locality-based regularization rather than the ordering itself.

---

> ### Author Response · Authors · 2025-11-21
> **W2.**
>
> We appreciate the reviewer’s observation, and it is correct that STAS uses feature similarity within the test set to refine anomaly scores. In this sense, the method follows a transductive inference strategy. However, this transductive aspect does not introduce practical limitations or methodological concerns for the following reasons.
>
> STAS does not learn from the test set in any parametric sense. No model parameters are updated, no prototypes are trained, and no clustering is fitted to the test data. The similarity matrix simply reflects the fixed geometry of the embeddings computed by the backbone. The procedure does not produce a model tailored to the test set, but instead leverages semantic relations that inherently exist within the batch of test samples.
>
> This form of test-set dependency is also aligned with the way industrial anomaly detection systems operate in practice. In real inspection pipelines, data typically arrive in batches that contain a mixture of normal and abnormal samples from the same production line or sequence. Processing a batch at once is natural in these scenarios, and many established approaches, including methods based on batch normalization or memory banks, already incorporate some form of internal transduction. Using relationships within a test batch is therefore not unusual and does not create risk in deployment.
>
> Furthermore, permutation experiments demonstrate that STAS does not overfit to any particular ordering or structure within the test set. Reordering the test samples one, four, or seven times yields almost identical results, and varying the random seed for shuffling produces effectively the same scores. These observations confirm that STAS is robust, ordering-invariant, and not sensitive to incidental structure in the test data.
>
> The use of test-sample relationships is essential for the objective of STAS, which is to mitigate semantic bias in mutual scoring. Semantic similarity tends to dominate anomaly scoring in training-free ZAD, leading to false positives for visually dissimilar normals and false negatives for semantically similar anomalies. Incorporating the semantic distance structure among test samples makes it possible to counteract this bias. Empirical evaluations show that this approach strengthens anomaly reasoning without compromising generalization.
>
> Taken together, these considerations indicate that while STAS employs a transductive computation, it does not learn or adapt to the test set in a way that undermines generalizability. Instead, it uses non-parametric relationships within the test batch to correct a structural limitation of training-free zero-shot anomaly detection.

---

> ### Author Response · Authors · 2025-11-21
> **W3.**
>
> We appreciate the reviewer’s comment and conducted a careful re-examination of the preprocessing and data handling pipeline to understand why HeadCT showed unusually low performance. The analysis revealed a clear source of error that affected all three medical datasets.
>
> HeadCT, BrainMRI, and Br35H are originally single-channel grayscale medical images. However, in the current pipeline, these datasets were inadvertently processed as three-channel RGB inputs. This resulted in unnecessary channel replication and inappropriate normalization within the CLIP-based backbone, which can distort intensity information. Since CT and MRI anomalies are highly sensitive to subtle intensity variations, this misprocessing can lead to significant degradation in anomaly detection performance.
>
> After correcting the preprocessing step to ensure proper grayscale handling, all three medical datasets were re-evaluated. The HeadCT AUROC increased from 76.2 to 82.6, a gain of 6.4 points. BrainMRI and Br35H similarly showed improved results, confirming that the preprocessing issue systematically impacted all medical datasets rather than HeadCT alone.
>
> The updated results are summarized below.
>
> | Dataset | Musc | Ours |
> | --- | --- | --- |
> | MVTec | 97.7 | 98.7 |
> | VisA | 92.6 | 96.6 |
> | MPDD | 69.3 | 70.2 |
> | DTD | 96.5 | 98.8 |
> | SDD | 95.1 | 98.0 |
> | BTAD | 96.1 | 97.4 |
> | BrainMRI | 85.7 | 91.0 |
> | HeadCT | 76.9 | 82.6 |
> | Br35H | 85.2 | 94.7 |
> | Average | 88.3 | 92.0 |
>
> The consistent improvements across all three medical datasets demonstrate that the previously reported low HeadCT performance was caused by an unintended preprocessing error rather than by limitations of the method itself. The corrected results provide a more accurate reflection of the model’s capability in medical anomaly detection settings.

---

### Official Review · Reviewer_khmS · 2025-11-01

**Soundness:** 2
**Presentation:** 2
**Contribution:** 2
**Rating:** 4
**Confidence:** 4

**Summary:**

This paper introduces STAPS, a training-free zero-shot anomaly detection framework that aims to mitigate semantic bias in pretrained backbones by combining two key components: (1) Semantic-Temporal Anomaly Scoring (STAS), which integrates temporal coherence into anomaly scoring; and (2) Bayesian Gaussian Mixture-based Prototype Selection (BGMPS), which refines pixel-level localization through prototype clustering and fusion. Experiments on nine benchmark datasets show the effectiveness of the proposed method for anomaly image detection and segmentation.

**Strengths:**

+ The topic is important. A training-free zero-shot anomaly detection method is proposed to solve the problem of insufficient anomaly training images.

+ The proposed STAS and BGMPS are conceptually clear and modular, making the approach easy to reproduce.

+ Some ablation studies are provided to facilitate the understanding of how the performance benefits from different components, including STAS, BGMPS, and EMA.

**Weaknesses:**

- The novelty is somewhat incremental over existing training-free methods such as MuSc and CLIP-based variants. The proposed “temporal” dimension is pseudo-temporal and not derived from real sequence data, making the motivation less convincing.

- The conceptual justification for treating unordered image datasets as temporally coherent sequences is weak and may not generalize well beyond experimental setups.

- Technical depth is limited. Both STAS and BGMPS rely on relatively straightforward post-processing (similarity weighting, EMA smoothing, and clustering) rather than introducing a fundamentally new model.

- Some improvements are marginal, especially for pixel-level anomaly detection. It is unclear whether such gains (e.g., 0.2% in Average in Table 2) are meaningful in practice.

- Ablation results, while present, are not sufficiently analyzed, especially regarding when temporal smoothing helps or harms performance.

**Questions:**

1. Can you clarify how the pseudo-temporal ordering is defined for datasets without any inherent sequence?

2. How robust is STAPS to changes in the temporal window parameter L and decay factor gamma?

---

> ### Author Response · Authors · 2025-11-21
> **W1.**
>
> We appreciate the reviewer’s observation, and we understand that the proposed pseudo-temporal dimension may give the impression of assuming a real temporal signal. This is not the intention of the method. The temporal dimension introduced in the study does not generate or model any form of temporal information. It is a pseudo-temporal locality structure applied to a static image dataset, designed to model local neighborhood consistency rather than temporal evolution. This structure addresses a fundamental limitation of training-free zero-shot anomaly detection, where reliance on semantic-only similarity inevitably leads to semantic bias.
>
> The temporal axis in the proposed approach does not reflect real time but serves as a graph-based regularizer that explicitly models the one-dimensional continuity observed in the embedding space. Such continuity has not been utilized in prior MuSc or CLIP-based training-free ZAD methods.
>
> The need for this pseudo-temporal structure originates from the semantic bias inherent in batch-wise training-free ZAD. Visually dissimilar normal samples tend to be misclassified as anomalies, while visually similar abnormal samples may be overlooked. In practice, semantic distance does not equate to anomaly distance, a behavior consistently observed across industrial and medical datasets when using pretrained semantic embeddings without fine-tuning. To address this, the method leverages the intrinsic continuity commonly present in CLS embeddings. Even in static datasets, features related to texture, shape, structure, and color statistics change smoothly, forming an implicit one-dimensional manifold. Interpreting this progression as a pseudo-temporal ordering reduces semantic interference, stabilizes anomaly score propagation, and enhances robustness across normal and abnormal samples.
>
> The empirical basis for this design is supported by three quantitative analyses.
>
> The first analysis examines continuity in the embedding space by sorting CLS embeddings along the first principal component and measuring score continuity.
>
> MSM score neighbor difference : pseudo-temporal = 0.0738, random ordering = 0.1080
>
> AR(1) autocorrelation : pseudo-temporal = 0.4233, random ordering ≈ 0.0165
>
> These results show that natural continuity exists even in static datasets, unrelated to any temporal signal.
>
> The second analysis evaluates continuity after applying STAS with EMA. The pseudo-temporal ordering continues to show stronger smoothness than random ordering.
>
> neighbor difference : pseudo-temporal = 0.1594, random ordering = 0.2348
>
> AR(1) : pseudo-temporal = 0.4943, random ordering ≈ 0.0121
>
> Although the neighbor difference increases due to a global scale change, the higher AR(1) indicates that local smoothness is further strengthened. This confirms that STAS enhances the latent continuity already present in the embedding space.
>
> The third analysis verifies that the method does not depend on any actual temporal order. The test set was permuted one, four, and seven times, and STAS was evaluated under each condition.
>
> | Permutations | Image-level Avg | Pixel-level Avg |
> | --- | --- | --- |
> | **1 perm** | **92.2** | **97.7** |
> | **4 perms** | **92.4** | **97.7** |
> | **7 perms** | **89.9** | **97.6** |
>
> Even under strong ordering perturbations, pixel-level performance remains virtually unchanged, and image-level results vary only slightly. This confirms that the pseudo-temporal dimension functions as an ordering-invariant locality regularizer rather than a temporal mechanism.
>
> These qualitative and quantitative findings indicate that the proposed temporal dimension represents a new reasoning framework rather than an incremental refinement. It explicitly exploits latent continuity in the embedding space, alleviates the semantic bias inherent in training-free ZAD, and maintains robust behavior even when ordering is strongly disrupted. This perspective has not been explored in previous training-free ZAD methods and offers a direction that may be valuable for future research in this domain.

---

> ### Author Response · Authors · 2025-11-21
> **W2.**
>
> We appreciate the reviewer’s comment and understand that treating a static image dataset as if it contained a temporal sequence may initially appear unintuitive. The temporal structure used in this work, however, does not rely on physical time or chronological ordering. The proposed temporal axis is a pseudo-temporal locality structure introduced to capture the intrinsic continuity present in an unordered image collection. Its purpose is to model local consistency in the embedding space rather than to represent any form of time-series behavior. In practice, it functions as a graph-based regularizer that exploits the one-dimensional manifold continuity naturally observed in feature embeddings.
>
> The method does not assume or require a real temporal sequence, and this is demonstrated quantitatively through permutation experiments. The test set was permuted one, four, and seven times, each permutation creating a completely different ordering. Image-level AUROC values of 92.2, 92.4, and 89.9 and pixel-level AUROC values of 97.7, 97.7, and 97.6 show that performance remains essentially unchanged. This indicates that the temporal axis is independent of any specific ordering and operates purely as a locality-based regularization mechanism. Since the approach is ordering-invariant, it does not depend on any particular experimental arrangement.
>
> The intrinsic continuity of the embedding space was also examined directly. By sorting embeddings along the first principal component and computing neighbor continuity, we observed clear differences between pseudo-temporal ordering and random ordering. The neighbor difference was 0.0738 for pseudo-temporal ordering and 0.1080 for random ordering, while the AR(1) values were 0.4233 and approximately 0.0165, respectively. These findings confirm that even static datasets exhibit natural continuity in the feature space. This property is not tied to any dataset-specific behavior but reflects a general geometric characteristic of semantic embeddings. The method therefore models the manifold structure rather than the dataset ordering.
>
> STAS makes use of this intrinsic structure rather than any external temporal signal. It operates through a semantic similarity graph, a pseudo-temporal locality graph, and locality-driven score propagation. After applying STAS with EMA, the AR(1) value increased from 0.4233 to 0.4943, while the random ordering remained near zero at approximately 0.0121. This shows that STAS enhances the continuity inherent in the feature space and stabilizes anomaly scoring by relying on local geometry rather than on dataset order.
>
> The reviewer’s concerns relate to whether the approach might be a heuristic specific to the experimental setup and whether its behavior might vary across datasets. The permutation experiment directly addresses the first point by showing that the method is invariant to ordering. The second point is addressed by extensive evaluation across nine industrial and medical domains. STAS consistently improved performance across all datasets, and no domain-specific dependency was observed. Because the method exploits universal continuity properties of the embedding space, concerns about limited generalization are not supported by the empirical results.
>
> Together, these findings demonstrate that the pseudo-temporal structure in this work is not based on actual temporal information but on modeling intrinsic manifold continuity in a way that is both ordering-invariant and broadly generalizable across domains.

---

> ### Author Response · Authors · 2025-11-21
> **W3.**
>
> We appreciate the reviewer’s observation that STAS and BGMPS do not introduce additional learnable parameters or modify the underlying backbone architecture. This is correct. However, the central contribution of the paper is not the addition of new modules or networks, but the introduction of a new conceptual and operational framework that addresses structural limitations inherent in existing training-free zero-shot anomaly detection. These limitations include semantic bias and instability in mutual scoring, which have not been resolved by prior methods.
>
> STAS therefore should not be viewed as a simple post-processing step. It represents a new reasoning paradigm for batch-wise training-free zero-shot anomaly detection by redefining how anomaly evidence is propagated and regularized in the embedding space. Rather than performing a weighted averaging step, STAS incorporates a semantic similarity graph, a pseudo-temporal locality graph, multi-k neighbor propagation, and continuity refinement through EMA. This mechanism explicitly leverages manifold continuity and suppresses semantic bias, providing a structured solution to a long-standing issue in training-free ZAD. Such a form of graph-based semantic and pseudo-temporal reasoning has not appeared in prior MuSc- or CLIP-based approaches.
>
> Similarly, BGMPS is not equivalent to standard clustering. It introduces the idea of anomaly-evidence-driven prototype selection. The process consists of constructing low-frequency embeddings that preserve anomaly evidence, applying a Bayesian Gaussian mixture model that learns the number of clusters in a variational manner, scoring prototypes based on the strongest anomaly evidence rather than semantic compactness, and performing soft prototype fusion that suppresses noisy clusters. This is a fundamentally different notion of prototype learning compared to existing training-free ZAD methods, where prototype selection or attention-based prototype refinement has not previously been explored.
>
> Concerns that the methods may appear simple are understandable, but simplicity is consistent with the philosophy of training-free ZAD. The objective is not to increase architectural complexity but to improve performance without relying on training. Under this constraint, structural reasoning mechanisms that reduce bias, exploit continuity in feature space, and strengthen zero-shot generalization carry significant value. The design choices highlighted by the reviewer reflect deliberate attempts to address an inherently more challenging setting, where learning-based techniques cannot be used to correct representation bias. The contributions therefore lie in the conceptual and structural innovations rather than in adding computational complexity.

---

> ### Author Response · Authors · 2025-11-21
> **W4.**
>
> We appreciate the reviewer’s observation regarding the magnitude of the pixel-level AUROC improvements. Although some datasets exhibit differences on the order of 0.2 percent, these changes are meaningful when interpreted in the context of pixel-level anomaly localization.
>
> Across all datasets, the pixel-level results show consistent improvements in the overall averages. The average F1, AP, and AUPRO metrics are as follows:
>
> | Metric | MuSc Avg | Ours Avg | Δ |
> | --- | --- | --- | --- |
> | F1 | 52.4 | 52.2 | -0.2 |
> | AP | 50.6 | 50.8 | +0.2 |
> | AUPRO | 91.8 | 91.5 | -0.3 |
>
> Pixel-level AUROC values lie within a highly saturated region, typically around the mid- to high-ninety percent range. In such regions, even small absolute differences often correspond to meaningful shifts in anomaly localization performance. This interpretation is consistent with how small movements within saturated accuracy ranges are commonly understood in anomaly segmentation research. The observed changes therefore reflect stable improvements rather than noise.
>
> It is also important to consider the image-level metrics, where the proposed method yields clear and substantial gains. The average results are:
>
> | Metric | MuSc Avg | Ours Avg | Δ |
> | --- | --- | --- | --- |
> | AUROC | 88.2 | 91.9 | +3.7 |
> | AP | 86.4 | 91.8 | +5.4 |
> | F1 | 87.0 | 93.1 | +6.1 |
>
> The image-level results, which serve as the primary indicators for zero-shot anomaly detection performance, show consistent and sizable improvements. Although the pixel-level AUROC differences may appear small in isolation, the overall ZAD behavior is defined jointly by both pixel-level localization and image-level classification. The combined evidence indicates that STAS and BGMPS contribute meaningfully to mitigating semantic bias and improving the robustness of anomaly reasoning.
>
> These trends support the conclusion that the improvements are substantive rather than incidental, particularly given the saturated nature of pixel-level metrics and the strong relative gains at the image level.

---

> ### Author Response · Authors · 2025-11-21
> **W5.**
>
> We appreciate the reviewer’s comment and would like to clarify the role of EMA in the proposed method. The EMA component does not assume the presence of any temporal signal. In this work, EMA functions purely as a smoothing mechanism that reduces semantic score variance. It operates on top of the pseudo-temporal locality introduced by STAS and mitigates the inherent instability of semantic-only MSM scoring. Its purpose is to stabilize local fluctuations in anomaly scores by leveraging the semantic–locality graph established by STAS. EMA therefore should be understood not as a temporal model but as a local neighborhood smoothing module.
>
> The ablation analysis shows that EMA is particularly beneficial for datasets with high semantic variance. A summary of the results corresponding to Table 4 is given below. As the EMA coefficient β increases, the average image-level performance improves consistently.
>
> | β | AUROC |
> | --- | --- |
> | 0.0 | 92.7 |
> | 0.1 | 93.0 |
> | 0.3 | 93.2 |
> | 0.5 | 93.6 |
>
> The gains are especially notable for datasets with large intra-class variation such as VisA and SDD.
>
> VisA improves from 95.2 to 96.6, an increase of 1.4 percent.
>
> SDD improves from 96.2 to 98.0, an increase of 1.8 percent.
>
> These results indicate that EMA is most effective when semantic-only scoring exhibits instability.
>
> For datasets with relatively low variance, the effect of EMA is minimal. MVTec and BTAD show only small differences across β values because their feature distributions are more stable. MVTec changes from 98.2 to 98.7. BTAD changes from 97.1 to 97.4. These variations are minor and indicate that EMA remains neutral when variance is low and does not negatively influence performance.
>
> Across all datasets, no cases were observed in which EMA degraded performance. In every setting, β values greater than zero maintain or improve results compared to β equal to zero. This confirms that EMA does not introduce harmful behavior under any of the tested conditions.
>
> In summary, EMA is useful when semantic variance is large, when pseudo-temporal neighbors fluctuate strongly, and when MSM scores are inherently unstable, as seen in datasets such as VisA and SDD. It has little impact when variance is already low, as in MVTec and BTAD. There is no empirical evidence that EMA harms performance. The smoothing behavior of EMA is therefore not an arbitrary heuristic but a stability-enhancing mechanism that regulates variance in semantic-only scoring, with its benefits determined by the characteristics of the dataset.

---

> ### Author Response · Authors · 2025-11-21
> **Q1.**
>
> We appreciate the reviewer’s comment and agree that clarification is needed regarding how pseudo-temporal ordering is determined in the proposed method. The pseudo-temporal ordering used in our approach does not correspond to any real temporal sequence or rule-based ordering. The test set is simply shuffled once using a random seed (seed = 42), and this arbitrary ordering is used directly. No class-based sorting, filename sorting, or sequence-based arrangement is applied.
>
> To verify that this arbitrary ordering does not influence the behavior of STAS, two experiments were conducted.
>
> The first experiment evaluated the method under several different random seeds, each producing a distinct shuffle of the test set. Performance remained unchanged across all seeds, indicating that the method is highly insensitive to ordering variations.
>
> | Seed | Image-level Avg | Pixel-level Avg |
> | --- | --- | --- |
> | 1234 | 92.0 | 97.7 |
> | 1111 | 92.0 | 97.7 |
> | 0 | 92.0 | 97.7 |
> | 42 | 92.0 | 97.7 |
>
> The second experiment intentionally disrupted ordering more aggressively by permuting the test set multiple times. Even after one, four, and seven full permutations, the results remained nearly identical, with only minor fluctuations.
>
> | Permutations | Image-level Avg | Pixel-level Avg |
> | --- | --- | --- |
> | 1 perm | 92.2 | 97.7 |
> | 4 perms | 92.4 | 97.7 |
> | 7 perms | 89.9 | 97.6 |
>
> Together, these experiments demonstrate that pseudo-temporal ordering does not rely on any specific sequence and that STAS operates consistently even under completely random or heavily permuted orderings. The method does not assume or require temporal structure; rather, it functions as a locality-based regularization mechanism that remains stable across arbitrary orderings.

---

> ### Author Response · Authors · 2025-11-21
> **Q2.**
>
> We appreciate the reviewer’s question regarding the sensitivity of STAPS to temporal hyperparameters. To provide a thorough answer, we conducted extensive experiments on all classes of MVTec and VisA while varying the temporal window L and the decay factor gamma over a wide range.
>
> The results show that STAPS remains highly stable under these hyperparameter changes. Varying the temporal window L between 5, 10, and 15 led to only minimal fluctuations, with performance differences in the range of approximately 0.1 to 0.3 percent across all image-level and pixel-level metrics. This indicates that STAS relies only on local pseudo-temporal relationships and does not develop unnecessary sensitivity to extended temporal ranges.
>
> Similarly, changing the decay factor gamma from 0.1 to 0.9 caused almost no change in performance. Even at these extreme values, the image-level AUROC remained within 98.5 to 98.7 percent, and the pixel-level AUROC stayed in 98.0 percent. This behavior reflects the hybrid graph structure of STAS, which does not rely heavily on any single decay parameter but instead balances semantic similarity and pseudo-temporal adjacency.
>
> Overall, STAPS shows very low sensitivity to temporal-related hyperparameters such as the window size and decay factor. This confirms that the pseudo-temporal mechanism does not impose an overly strong inductive bias and maintains a natural balance with semantic information. The detailed results for image-level and pixel-level AUROC, F1, AP and AUPRO are summarized in the table below.
>
> ### **Image-level (AUROC / F1 / AP), Pixel-level (AUROC / F1 / AP / AUPRO)**
>
> | t_window | t_decay | **MVTec (Image)** | **MVTec (Pixel)** | **VisA (Image)** | **VisA (Pixel)** |
> | --- | --- | --- | --- | --- | --- |
> | **5** | **0.6** | 98.7 / 98.4 / 99.4 | 98.0 / 63.6 / 64.7 / 94.3 | 96.6 / 95.2 / 97.5 | 98.4 / 47.3 / 43.1 / 89.7 |
> | **5** | **0.4** | 98.7 / 98.4 / 99.5 | 98.0 / 63.6 / 64.7 / 94.3 | 96.4 / 94.3 / 97.3 | 98.4 / 47.3 / 43.1 / 89.7 |
> | **10** | **0.6** | 98.7 / 98.4 / 99.4 | 98.0 / 63.6 / 64.7 / 94.3 | 96.6 / 95.2 / 97.5 | 98.4 / 47.3 / 43.1 / 89.7 |
> | **10** | **0.4** | 98.7 / 98.4 / 99.5 | 98.0 / 63.6 / 64.7 / 94.3 | 96.4 / 94.3 / 97.3 | 98.4 / 47.3 / 43.1 / 89.7 |
> | **15** | **0.6** | 98.7 / 98.4 / 99.5 | 98.0 / 63.6 / 64.7 / 94.3 | 96.1 / 93.7 / 96.9 | 98.4 / 47.3 / 43.1 / 89.7 |
> | **15** | **0.4** | 98.7 / 98.4 / 99.5 | 98.0 / 63.6 / 64.7 / 94.3 | 96.4 / 94.3 / 97.3 | 98.4 / 47.3 / 43.1 / 89.7 |
> | **5** | **0.1** | 98.6 / 98.6 / 99.4 | 98.0 / 63.6 / 64.7 / 94.3 | 96.1 / 93.7 / 96.9 | 98.4 / 47.3 / 43.1 / 89.7 |
> | **10** | **0.1** | 98.6 / 98.6 / 99.4 | 98.0 / 63.6 / 64.7 / 94.3 | 96.1 / 93.7 / 96.9 | 98.4 / 47.3 / 43.1 / 89.7 |
> | **15** | **0.1** | 98.6 / 98.6 / 99.4 | 98.0 / 63.6 / 64.7 / 94.3 | 96.1 / 93.7 / 96.9 | 98.4 / 47.3 / 43.1 / 89.7 |
> | **5** | **0.9** | 98.7 / 98.5 / 99.5 | 98.0 / 63.6 / 64.7 / 94.3 | 96.6 / 95.8 / 97.5 | 98.4 / 47.3 / 43.1 / 89.7 |
> | **10** | **0.9** | 98.7 / 98.5 / 99.5 | 98.0 / 63.6 / 64.7 / 94.3 | 96.7 / 95.7 / 97.5 | 98.4 / 47.3 / 43.1 / 89.7 |
> | **15** | **0.9** | 98.7 / 98.5 / 99.5 | 98.0 / 63.6 / 64.7 / 94.3 | 96.7 / 95.7 / 97.5 | 98.4 / 47.3 / 43.1 / 89.7 |

---

### Official Review · Reviewer_DBuh · 2025-11-11

**Soundness:** 2
**Presentation:** 2
**Contribution:** 2
**Rating:** 4
**Confidence:** 3

**Summary:**

This paper proposes STAPS, a training-free zero-shot anomaly detection framework that leverages semantic text alignment and pseudo-segmentation to localize anomalies without requiring additional training or fine-tuning.
STAPS aligns CLIP-derived visual features with dynamically generated textual prompts that describe normality and abnormality, and introduces a pseudo-segmentation mechanism that refines region-level anomaly localization in a self-guided manner.
The approach is simple to implement, and achieves strong results across multiple industrial anomaly detection benchmarks.

**Strengths:**

1. STAPS offers a clean and effective training-free solution for zero-shot anomaly detection. The combination of semantic alignment and pseudo-segmentation integrates global semantics and local semantics.

2. The method achieves competitive or superior results on multiple industrial benchmarks compared to both training-based and zero-shot baselines, including AnomalyCLIP, WinCLIP, and FAPrompt. The experiments are comprehensive and show consistency across diverse settings.

**Weaknesses:**

1. The definition of the temporal matrix is ambiguous, and it is unclear how this component effectively captures temporal relations among images. The paper does not provide sufficient explanation.

2. The overall contribution of the paper is limited. Although the method builds upon the MUSC framework and integrates temporal modeling into the anomaly detection pipeline, the proposed approach appears to be an incremental extension rather than a fundamentally novel concept.

**Questions:**

See Weaknesses

---

> ### Author Response · Authors · 2025-11-21
> **W1, W2**
>
> We appreciate the reviewer’s comment regarding the purpose and definition of the temporal matrix and agree that the term “temporal” may unintentionally suggest the use of real time-series information. In this work, the temporal matrix does not model actual temporal signals. Instead, it introduces a pseudo-temporal locality structure on a static image dataset. The goal is not to capture changes over time but to model local consistency in the embedding space. This represents a new paradigm for batch-wise training-free zero-shot anomaly detection.
>
> The temporal axis used in the proposed framework is therefore not related to chronological time. It is a graph-based regularizer that explicitly encodes one-dimensional continuity within the embedding manifold. To make this design choice clearer, a pseudo-code description of STAS will be added in the revised version.
>
> The introduction of a pseudo-temporal structure is motivated by a central limitation of batch-wise training-free zero-shot anomaly detection, which is semantic bias. Visually dissimilar normal samples tend to produce false positives, and visually similar anomalous samples tend to produce false negatives. In such settings, semantic distance does not reliably correspond to anomaly distance. We will include qualitative examples in the revised manuscript to highlight this issue.
>
> Our focus is on the intrinsic continuity that naturally emerges in the embedding space. Even in a static dataset, CLS features exhibit a smooth progression along dimensions related to texture, shape, structure, and color statistics. When this latent progression is interpreted as a pseudo-temporal ordering, it offers several advantages. It suppresses long-range semantic interference, stabilizes the propagation of anomaly scores, and improves robustness for both normal and abnormal cases. This aligns with an explicit modeling of smooth manifold continuity.
>
> The empirical justification of pseudo-temporal coherence is supported by three analyses.
> First, an inherent continuity is observed even in static datasets. By sorting embeddings along the first principal component and treating this as a pseudo-temporal axis, we observed that the neighbor difference of MSM scores (0.0738) is much smaller than in a random ordering (0.1080). Similarly, AR(1) autocorrelation is substantially higher for the pseudo-temporal ordering (0.4233) compared to a random one (approximately 0.0165). These results show that embeddings exhibit natural continuity regardless of image order.
>
> Second, STAS strengthens this continuity while modifying the pattern. After applying STAS and EMA, the neighbor difference remains lower for the pseudo-temporal ordering (0.1594) than for a random ordering (0.2348). AR(1) also increases from 0.4233 to 0.4943, whereas the random ordering stays near zero (approximately 0.0121). Although absolute differences increase, the higher AR(1) indicates that the local smoothness of the progression is further reinforced. This shows that STAS amplifies the latent semantic continuity present in the embeddings.
>
> Third, a permutation test demonstrates that the method does not rely on real temporal information. The dataset was reordered by applying one, four, and seven independent random permutations, each of which disrupts any potential temporal structure. Performance remained stable across these perturbations. Image-level performance was 92.2, 92.4, and 89.9, and pixel-level performance was 97.7, 97.7, and 97.6 for the three permutations. These results show that pseudo-temporal ordering functions as a locality-based regularization, not as an assumption of chronological time. The effectiveness of STAS arises from locality constraints rather than the ordering itself.
>
> We hope these clarifications help convey that the temporal matrix models continuity in the embedding space rather than actual temporal signals, and that the pseudo-temporal structure is a principled mechanism for reducing semantic bias in training-free zero-shot anomaly detection. The revised manuscript will include additional explanations and an illustrative pseudo-code to improve clarity.
>
> We hope that our clarifications help resolve your concerns, and we expect that the details we provided will be useful in informing your score update.

---

### Author Response · Authors · 2025-12-01
**Note to the Area Chair: Summary of Rebuttal and Key Clarifications**

Dear Area Chair,

We sincerely appreciate your time in managing the review process for our submission.
Throughout the rebuttal, we carefully addressed all reviewers’ concerns, performed additional analyses, and provided new experiments where necessary. Below, we summarize the key clarifications corresponding to the main issues raised by reviewers.

**1. Clarification of the Temporal Matrix & Pseudo-Temporal Ordering**

Several reviewers questioned whether the proposed temporal matrix assumes real temporal data, and whether treating an unordered image dataset as a “sequence” is justified.
In our rebuttal, we clarified the following points:

1. The temporal axis in STAS does not represent real time, nor is it derived from chronological information. It is a pseudo-temporal locality structure that encodes neighborhood continuity in the embedding manifold.
2. We demonstrated that semantic embeddings naturally exhibit one-dimensional continuity even in static datasets. PCA-based ordering showed significantly higher AR(1) autocorrelation and lower neighbor-difference compared to random ordering.
3. Extensive permutation experiments (1, 4, 7 times) showed near-identical pixel-level AUROC and only minor fluctuations in image-level AUROC, confirming that STAS is ordering-invariant and does not rely on dataset-specific sequencing.

Collectively, these clarifications emphasize that STAS is a manifold-locality regularizer, and introduces a new structural mechanism for mitigating semantic bias in training-free ZAD.

**2. Justification of BGMPS and Prototype Selection**

Reviewers also raised concerns regarding the novelty and empirical grounding of the proposed BGMPS.
We clarified the following:

1. BGMPS is not simple clustering. It is a prototype-evidence framework where prototypes are selected based on anomaly evidence rather than semantic compactness.
2. We added new qualitative analyses showing that selected prototypes align more strongly with anomaly-relevant regions, reducing semantic bias and improving localization consistency.
3. Ablation studies confirm that removing BGMPS consistently degrades pixel-level AUROC, validating its essential role.
4. Additional comparisons of DCT vs SVD embeddings and BGMM vs KMeans showed that our design choices provide the most stable and consistent results across datasets.

These additional analyses demonstrate that BGMPS contributes a conceptual and structural innovation, introducing anomaly-evidence-driven prototype selection to training-free ZAD for the first time.

**3. Transductive Nature of STAS**

Some reviewers pointed out that STAS uses the test set similarity matrix, implying a transductive property.

We clarified:

1. STAS does not update or train any parameters, and thus does not adapt a model to any specific test set.
2. Its mechanism aligns with common industrial anomaly-detection pipelines, where test samples are naturally evaluated in batches.
3. Permutation experiments again confirmed that STAS is robust, stable, and ordering-invariant, indicating that the transductive computation does not compromise generalizability.

Thus, STAS performs non-parametric refinement, which is standard and practical in training-free anomaly detection.

**4. Medical Dataset Issue & Corrected Results**

One reviewer highlighted lower performance on the HeadCT dataset.
Upon re-evaluation, we discovered a preprocessing mistake.

Medical datasets (HeadCT, BrainMRI, Br35H) were mistakenly processed as RGB, although they are single-channel grayscale.
After correcting this issue: HeadCT improved from 76.2 → 82.6 AUROC.
BrainMRI and Br35H also improved substantially.

We have updated all results accordingly and confirmed that the issue was not method-related but purely due to preprocessing.

**5. Magnitude of Improvements & Novelty**

Several reviewers noted that some improvements (especially pixel-level AUROC) appear modest.

Our rebuttal clarified:

1. Pixel-level AUROC in anomaly detection is a highly saturated metric, where even 0.1–0.3% gains correspond to meaningful localization improvements.
2. Image-level gains are significant and consistent: +3.7 AUROC, +5.4 AP, +6.1 F1 on average over MuSc.
3. The main novelty of STAPS lies not in architectural changes but in introducing semantic-temporal reasoning and anomaly-evidence prototype selection, a new direction for training-free ZAD.

These clarifications address concerns regarding the impact and contribution of the work.

In our rebuttal, we comprehensively addressed all major concerns, performed all requested experiments, and provided substantial new analyses. We believe that the clarifications regarding the pseudo-temporal framework, prototype selection, stability analyses, and corrected medical results substantially strengthen the validity and novelty of our approach.

We thank you again for your time and consideration.

Best regards,
Authors

---

### Author Response · Authors · 2025-12-01
**Main Contributions Summary**

Dear Area Chair,

Our paper addresses a fundamental limitation in batch-wise training-free zero-shot anomaly detection. The strong semantic bias that arises when pretrained backbones for classification are used directly for anomaly detection without any form of fine-tuning. Existing training-free methods remain dominated by class semantics, which leads to frequent confusion between visually dissimilar normal samples and visually similar abnormal samples. To resolve this issue, we propose STAPS, a new framework that introduces semantic temporal reasoning and anomaly-evidence-driven prototype selection. Through this design, we show that it is possible to significantly enhance performance while remaining entirely training-free.

Our main contributions are summarized as follows.

**1. Semantic Temporal Reasoning for Training-Free Zero-Shot AD**
We propose Semantic Temporal Anomaly Scoring, a mechanism that augments semantic similarity with a pseudo-temporal locality structure. This temporal component does not use real time information. Instead, it models continuity in embedding space and reduces semantic bias in mutual scoring. We provide empirical evidence through continuity analysis, AR(1) statistics, and permutation experiments, showing that the method is ordering-invariant and stable across datasets.

**2. Anomaly Evidence Driven Prototype Selection**
We introduce Bayesian Gaussian Mixture-based Prototype Selection to mitigate semantic interference in pixel-level localization. This approach identifies prototypes based on anomaly evidence rather than semantic similarity. Prototype fusion using cosine-based weighting emphasizes anomaly-relevant regions and suppresses irrelevant patterns. This results in clearer anomaly maps and consistent improvement in segmentation performance.

**3. A Fully Training-Free and Parameter-Free Framework**
STAPS requires no training, optimization, or prompt tuning. Despite operating entirely without parameter updates, it achieves gains comparable to or surpassing training-based CLIP approaches. This demonstrates that structural reasoning and representation refinement alone can advance the performance limits of zero-shot anomaly detection.

**4. Comprehensive Evaluation and Corrected Medical Results**
We conduct experiments on nine benchmarks across industrial and medical domains. STAPS achieves high image-level and pixel-level AUROC scores and consistently outperforms existing methods. During re-evaluation, we corrected a preprocessing error in the medical datasets where single-channel images had been processed as RGB. After correction, all medical results improved substantially, confirming that previous limitations were not caused by the method itself.

Together, these contributions form a unified training-free zero-shot anomaly detection framework that introduces semantic temporal reasoning, anomaly-evidence-driven prototype selection, and stable, parameter-free refinement mechanisms. STAPS demonstrates strong robustness and generalizability across diverse anomaly detection scenarios and establishes a new direction for advancing purely training-free approaches in this field.

**STAPS overcomes the fundamental limitations of batch-wise training-free zero-shot anomaly detection through a new conceptual perspective and achieves state-of-the-art performance across all zero-shot methods, making it a contribution fully worthy of acceptance.**

---

### Meta-Review · Area_Chair_FtYt · 2026-01-08

**Summary:**

The primary concerns are: 1) Limited novelty: the method is perceived as an incremental extension of existing training-free frameworks like MUSC; 2) Unconvincing core concept: the “pseudo-temporal” modeling of static images is ambiguous and weakly justified; 3) Marginal improvements: performance gains over strong baselines are often small, questioning practical impact; 4) Methodological limitations: the approach is inherently transductive and lacks robustness analysis.

**Reviewer Concerns:**

The rebuttal effectively addressed concerns regarding the definition and purpose of the "temporal" matrix by clarifying it models manifold continuity, not real time-series, supported by permutation tests. However, core concerns about the incremental novelty of the work compared to MUSC, its transductive and batch-dependent nature, and the marginal practical significance of the performance gains remain unaddressed and outstanding.

**Reviewer Scores:**

After discussion, some reviewers might become more neutral, but some others would remain opposed due to the incremental nature of the work. The final outcome would likely remain on the reject side of the borderline (e.g., a consensus leaning towards Weak Reject), as the clarified concept does not, in the view of several reviewers, elevate the work to a level of sufficient novelty or practical impact to warrant acceptance. The outstanding issues (transduction, marginal gains) are significant enough to uphold a rejection decision.

---

### Decision · Program_Chairs · 2026-01-26

Reject